# SM9 Identity-Based Encryption with Designated-Position Fuzzy Equality Test

Siyue Dong [1,*,†], Zhen Zhao [1,*,†], Baocang Wang [1], Wen Gao [2] and Shanshan Zhang [3]

1 The State Key Laboratory of Integrated Service Networks, Xidian University, Xi'an 710071, China; bcwang@xidian.edu.cn
2 School of Cyberspace Security, Xi'an University of Posts & Telecommunications, Xi'an 710121, China; gaowen@xupt.edu.cn
3 School of Mathematics and Information Science, Baoji University of Arts and Science, Baoji 721016, China; sszhang@bjwlxy.edu.cn
* Correspondence: sydong@stu.xidian.edu.cn (S.D.); zzhen@xidian.edu.cn (Z.Z.)
† These authors contributed equally to this work.

**Abstract:** Public key encryption with equality test (PKEET) is a cryptographic primitive that enables a tester to determine whether two ciphertexts encrypted with same or different public keys have been generated from the same message without decryption. Previous studies extended PKEET to public key encryption with designated-position fuzzy equality test (PKE-DFET), enabling testers to verify whether plaintexts corresponding to two ciphertexts are equal while ignoring specific bits at designated positions. In this work, we have filled the research gap in the identity-based encryption (IBE) cryptosystems for this primitive. Furthermore, although our authorization method is the all-or-nothing (AoN) type, it overcomes the shortcomings present in the majority of AoN-type authorization schemes. In our scheme, equality tests can only be performed between a ciphertext and a given plaintext. Specifically, even if a tester acquires multiple AoN-type authorizations, it cannot conduct unpermitted equality tests between users. This significantly reduces the risk of user privacy leaks when handling sensitive information in certain scenarios, while still retaining the flexible and simple characteristics of AoN-type authorizations. We use the Chinese national cryptography standard SM9-IBE algorithm to provide the concrete construction of our scheme, enhancing the usability and security of our scheme, while making deployment more convenient. Finally, we prove that our scheme achieves F-OW-ID-CCA security when the adversary has the trapdoor of the challenge ciphertext, and achieves IND-ID-CCA security when the adversary does not have the trapdoor of the challenge ciphertext.

**Keywords:** public key encryption with equality test; identity-based cryptography; designated-position fuzzy equality test

## 1. Introduction

The emergence of cloud computing has shifted a significant computational and storage burden from users to cloud servers, resulting in a continuous decrease in the cost of data processing and storage. As a result, there is a surge of cloud-based applications, such as the Internet of Things, big data, and artificial intelligence, which further propel the advancement of cloud technology [1–4]. Compared to traditional local storage systems, cloud storage provides several advantages including cost-effectiveness, scalability, easy management, and maintenance. With cloud technology, users have the flexibility to select storage capacity and service types that align with their specific needs. Additionally, they can seamlessly carry out operations such as data uploading, downloading, backup, archiving, and sharing. However, a cloud computing setting often implies multiple users share hardware. In light of security concerns, data on a cloud is usually in an encrypted form.

Consequently, there is an urgent need for novel cryptographic primitives tailored for processing encrypted data on cloud servers.

Several different cryptographic primitives are proposed to address the operation on encrypted data, such as searchable encryption [5,6], fully homomorphic encryption [7], and public key encryption with keyword search (PEKS) [8]. In PEKS, the server can check whether an encrypted ciphertext $C$ is derived from a plaintext $M$ without decryption. This property makes it well-suited for applications like the classification of public-key ciphertexts.

The above-mentioned PEKS scheme shares a similar limitation with previous searchable encryption schemes: it can only support operations on ciphertexts derived from the same public key, making it very limited in scenarios involving ciphertexts from multiple users. In 2010, Yang et al. [9] proposed a new cryptographic primitive: public key encryption with equality test (PKEET). In PKEET, an entity can check whether the corresponding plaintexts of two different ciphertexts encrypted with distinct public keys are equal without decryption.

In the original PKEET scheme [9], any entity can test the equality on two different users' ciphertexts. Undoubtedly, this presents a significant risk to the confidentiality of the ciphertext. Therefore, researchers continuously supplemented and extended the notion of PKEET afterward. Their works started with restricting the authority of the tester. Tang proposed two different authorization approaches, denoted as the fine-grained PKEET (FG-PKEET) [10] and all-or-nothing PKEET (AoN-PKEET) [11]. In the former, two users need to jointly negotiate a token for the tester before the tester can perform an equality test on the ciphertexts of these two users. The advantage of this approach is that it can effectively limit the tester from conducting unauthorized equality tests between users. However, the drawback is that users need to jointly negotiate tokens online for authorization, and the tester also needs to store a lot of tokens for each user. If there are $n$ users, the tester needs to store $\frac{n(n-1)}{2}$ tokens. In AoN-PKEET, each user can independently generate its own token for authorization to the tester, who can then perform equality test among users who have submitted their tokens. Therefore, only $n$ tokens need to be stored for $n$ users. However, the drawback is that if user A only wants to perform ciphertext equality test with B, after having received A's token, the tester can still perform equality test between the ciphertexts of A and those of user C who also authorized the tester with its token. Later, Huang et al. [12] proposed another authorization approach, which includes not only the approach in AoN-PKEET, but also an approach in which users can issue tokens on specific ciphertexts, and the tester cannot perform equality test on ciphertexts other than those authorized by the users. Building on these works, Ma et al. [13] proposed PKEET supporting flexible authorization (PKEET-FA). In this work, various authorization approaches were thoroughly summarized, essentially covering and integrating all types of authorization in previously proposed PKEET schemes.

We propose the notion of identity-based encryption with designated-position fuzzy equality test (IBE-DFET) and construct a concrete SM9-IBE-DFET scheme in this paper. The contributions of this paper are as follows:

- We introduced the designated fuzzy equality test feature into IBEET and obtained an IBE-DFET scheme. In our construction, a tester can select a wildcard set, and after obtaining the authorized trapdoor from the user, it determines whether the plaintext underlying a ciphertext and the given plaintext are fuzzily equal while ignoring the designated positions in the wildcard set.
- Our scheme has a distinct advantage compared to other PKEET schemes, which is that although the authorization approach in our scheme is an all-or-nothing (AoN)-type authorization, i.e., the user personally authorizes the tester to perform an equality test on its ciphertext, the tester can only perform a fuzzy equality test on the ciphertext of that user and its own plaintext, but cannot equality test the ciphertext of that user and another user who also authorized the tester with its trapdoor. This undoubtedly greatly enhances the confidentiality of user's ciphertexts. In other words, our

scheme combines the flexible and convenient characteristics of AoN-type authorization while avoiding the drawback of testers obtaining additional information with AoN-type authorization.

- We use the Chinese national cryptography standard SM9-IBE algorithm [14] to construct the concrete scheme, with the symmetric encryption part using the Chinese national cryptography standard SM4 symmetric encryption algorithm. The use of standard algorithms demonstrates the practicality and security of our scheme in a very intuitive way. Specifically, the SM9-IBE algorithm based on the elliptic curve discrete logarithm problem not only has high security, but also has highly efficient bilinear pairing operations that are very suitable for constructing equality test schemes that still heavily rely on bilinear pairings.

## 2. Related Work

**Identity-Based Encryption with Equality Test (IBEET):** In 2016, Ma [15] extended PKEET to IBE [16], and constructed an IBEET scheme. In the same year, a similar IBEET scheme was proposed by Lee et al. [17]. Compared to PKEET, the IBEET scheme inherits the advantages of the IBE scheme: each user's public key is the identifier for that user, which greatly simplifies key management and makes system deployment more convenient. Due to the favorable properties of IBEET, it has become a research hotspot, and a large number of related articles have subsequently appeared [18–28]. It is worth mentioning that some researchers have extended PKEET to other ID-based cryptosystems, resulting in attribute-based encryption with equality test (ABEET) [29–34] and certificateless encryption with equality test(CLEET) [35–37].

**Public Key Encryption with Fuzzy Matching (PKEFM):** Wang et al. [38] proposed the concept of Public Key Encryption with Fuzzy Matching (PKEFM), which can determine whether the edit distance between two encrypted messages is lower than a threshold. Thus, it can determine whether two messages are fuzzy equal based solely on their ciphertexts. This work is particularly suitable for detecting the equality of two messages when only a small number of foreseeable differences exist between them, such as spelling errors or differing formats. However, this scheme employs a method of computing similarity for equality test, making it unable to fulfill the task if users wish to check for fuzzy equality ignoring some designated positions.

**Public Key Encryption with Designated-Position Fuzzy Equality Test (PKE-DFET):** Zhao et al. [39] proposed a novel PKE-DFET scheme. Unlike most previous PKEET schemes, in this scheme, the tester can select a wildcard set, and when comparing two ciphertexts, the corresponding positions of the underlying plaintext indicated by the wildcard set will not affect the result of the equation test. In other words, the tester does not care whether the plaintexts at the wildcard set positions are equal or not. This scheme achieves fuzzy equation test at designated positions, making it highly suitable for constructing systems with specific requirements, i.e., systems with slight differences in plaintexts.

The DFET primitive has not yet been introduced into IBE cryptosystems where the certificate management problem is solved. Furthermore, in the original PKE-DFET scheme and many schemes supporting equality test for AoN-type authorizations, the tester can choose the object of equality test arbitrarily after obtaining authorization. In certain scenarios, such as dealing with highly sensitive medical records, the tester's not-permitted equality test could likely lead to the disclosure of patient's privacy. However, if we do not use the AoN-type authorization, for instance, if we use the FG-type [10] or ciphertext-level [13] authorization mentioned above, the complexity of the system will significantly increase. The above problems are well addressed in this work.

## 3. Preliminary

### 3.1. Bilinear Pairing

Let $\mathbb{G}_1$ and $\mathbb{G}_2$ be two additive cyclic groups of order $N$, and $\mathbb{G}_T$ be a multiplicative cyclic group of order $N$, where $N$ is a prime number. $P$ and $Q$ are generators of $\mathbb{G}_1$ and $\mathbb{G}_2$ respectively.

A bilinear pairing $e : \mathbb{G}_1 \times \mathbb{G}_2 \to \mathbb{G}_T$ satisfies the following properties:

(1) Bilinear: For any $P \in \mathbb{G}_1, Q \in \mathbb{G}_2, a, b \in \mathbb{Z}_N^*$, we have $e([a]P, [b]Q) = e(P, Q)^{ab}$.

(2) Non-degenerate: There exist elements $P \in \mathbb{G}_1, Q \in \mathbb{G}_2$, such that $e(P, Q) \neq 1_{\mathbb{G}_T}$, where $1_{\mathbb{G}_T}$ is the identity element of $\mathbb{G}_T$.

(3) Computable: For any $P \in \mathbb{G}_1, Q \in \mathbb{G}_2, e(P, Q)$ can be computed efficiently.

Let $P \in \mathbb{G}_1, Q \in \mathbb{G}_2$, the security of the bilinear pairing is mainly based on the computational difficulty of the following problem.

### 3.2. Decision Bilinear Inversion Diffie–Hellman (DBIDH) Assumption [40,41]

For any positive integers $a, b, r \in_R \mathbb{Z}_N^*$, it is hard to distinguish

$$\left( P_1, P_2, [a]P_i, [b]P_j, e(P_1, P_2)^{b/a} \right) \text{ and } \left( P_1, P_2, [a]P_i, [b]P_j, e(P_1, P_2)^r \right),$$

for some values of $i, j \in \{1, 2\}$.

### 3.3. Gap-$\tau$-Bilinear Collision Attack Assumption (Gap-$\tau$-BCAA1) [40,41]

For any positive integers $\tau, x \in_R \mathbb{Z}_N^*$, given

$$\left( P_1, P_2, [x]P_i, h_0, (h_1, [\frac{x}{h_1 + x}]P_j), \ldots, (h_\tau, [\frac{x}{h_\tau + x}]P_j) \right),$$

for some values of $i, j \in \{1, 2\}$, where $h_i \in_R \mathbb{Z}_N^*$ and different from each other for $0 \leq i \leq \tau$, and a DBIDH oracle which solves a given DBIDH problem, computing $e(P_1, P_2)^{x/(h_0 + x)}$ is hard.

### 3.4. Public Key Encryption with Designated Fuzzy Equality Test [39]

We start with DFET: if every bit in two messages is equal except for the designated *ignorable* positions, then we can say that these two messages are designated-position fuzzy equal. A set of the designated positions is defined as a *wildcard set*, which means the bit positions in this set do not affect the result of the equality test. For example, if the wildcard set is $\{3, 5\}$. Then the bit-string 10110 is designated-position fuzzy equal to 10011 but not to 00110.

The key technique of the proposed PKE-DFET is based on Viète formula [42,43], given two vectors $\vec{x} = \{x_1, x_2, \ldots, x_n\}, \vec{y} = \{y_1, y_2, \ldots, y_n\}$, and a set of wildcard $J = \{j_1, j_2, \cdots, j_m\} \subsetneq \{1, 2, \ldots, n\}$, the statement $x_i = y_i \vee i \in J$ for $i = \{1, 2, \ldots, n\}$ is the same as

$$\sum_{i=1}^{n} x_i \prod_{j \in J} (i - j) = \sum_{\substack{i=1 \\ i \notin J}}^{n} x_i \prod_{j \in J} (i - j) = \sum_{\substack{i=1 \\ i \notin J}}^{n} y_i \prod_{j \in J} (i - j) = \sum_{i=1}^{n} y_i \prod_{j \in J} (i - j).$$

Modify the equation with Viète formula [42,43], $\prod_{j \in J} (i - j) = \sum_{k=0}^{m} a_k i^k$, where $a_k$ is the coefficient of $i^k$, we have

$$\sum_{k=0}^{m} a_k \sum_{i=1}^{n} x_i i^k = \sum_{i=1}^{n} x_i \prod_{j \in J} (i - j) = \sum_{i=1}^{n} y_i \prod_{j \in J} (i - j) = \sum_{k=0}^{m} a_k \sum_{i=1}^{n} y_i i^k.$$

The definition of PKE-DFET is as follows: Given a wildcard set in PKE-DFET, the underlying messages of two ciphertexts in PKE-DFET will be regarded as designate-position fuzzy equal if they are equal on every position except for those belonging to the wildcard set.

Formally, given two ciphertexts $CT_1$ and $CT_2$, with corresponding plaintexts $M_1$ and $M_2$, we define a wildcard set $J = \{j_1, j_2, \cdots, j_m\} \subsetneq N$, where $N = \{1, 2, \cdots, n\}$. We represent messages in bit form as $M_i = M_{i,1}, M_{i,2}, \cdots, M_{i,n}$. If

$$M_{1,j} = M_{2,j} \text{ for each } j \in N \backslash J,$$

where $N \backslash J \cap J = \varnothing$ and $N \backslash J \cup J = N$, then we can say that $M_1$ and $M_2$ are designated-position fuzzy equal under the wildcard set $J$. This is denoted as

$$M_1 \backslash J = M_2 \backslash J.$$

It is evident that when $J = \varnothing$, this designated-position fuzzy equality test becomes a regular equality test. On the other hand, when $J = N$, the equality test becomes meaningless because any two ciphertexts would be fuzzy equal. However, to ensure the practicality of the scheme, $J$ must be freely selected by the test, but it should not be too large. Otherwise, for example, if we receive a ciphertext $CT_1$ generated from a plaintext $M_1$, and the tester encrypt another message $M_2$ of the same length to obtain $CT_2$ while selecting $J = \{2, 3, \cdots, n\}$, then test whether $M_1 \backslash J = M_2 \backslash J$ would directly reveal whether the first bit of the two messages is equal. By repeating this process, we can determine the value of message $M_1$ after $n$ tests. This renders the scheme insecure. Hence, we need to set a general upper bound $U$ and select the permissible wildcard set during encryption as $L$ where $L \leq U$. When choosing $J$, the tester must satisfy $J \leq L$; otherwise, the algorithm will abort and output $\perp$.

*3.5. Chinese National Cryptographic Standard SM9 [14]*

In 1984, Shamir [16] proposed the concept of identity-based encryption, where users can use their identity-related information, such as mobile numbers and e-mail addresses, as their public keys. This approach directly addresses the certificate management problem in traditional public key encryption cryptosystem. This excellent cryptographic primitive has seen significant development over the decades. Chinese State Cryptography Administration introduced the SM9 identity-based encryption standard [14]. SM9 includes a variety of algorithms such as digital signature algorithm, key exchange protocol, key encapsulation mechanism (KEM), and public key encryption (PKE) algorithm. Its applications have continued to evolve in the subsequent years.

We take SM9-IBE as the basic structure of our scheme. It is essentially a hybrid encryption consisting of SM9-KEM and a symmetrical encryption algorithm as data encapsulation mechanism (DEM).

A brief overview of the SM9-IBE algorithm is given as follows:

Setup($1^\lambda$): Taking as input a security parameter $\lambda$, the setup algorithm generates the public parameter

$$pp = \{\mathbb{G}_1, \mathbb{G}_2, \mathbb{G}_T, P_1, P_2, e, hid, H_1\},$$

where $\mathbb{G}_1$ and $\mathbb{G}_2$ are two additive cyclic groups of order $N$, and $P_1 \in \mathbb{G}_1$ and $P_2 \in \mathbb{G}_2$ are the generators of the two groups. $\mathbb{G}_T$ is a multiplicative group with order $N$. $e$ represents the bilinear pairing: $\mathbb{G}_1 \times \mathbb{G}_2 \rightarrow \mathbb{G}_T$. $hid$ is an identifier for private key generation functions. $H_1 : \{0,1\}^* \rightarrow \mathbb{Z}_N^*$ is a cryptographic hash function. Additionally, there are some auxiliary functions: MAC (Message Authentication Code) is a function for message authentication, KDF (Key Derivation Function) is a function for key generation, and a secure DEM algorithm. KDF : $\{0,1\}^* \rightarrow klen$, where $klen = 256$. Afterward, the Key Generation Center (KGC) randomly selects $k \in [1, N-1]$, and calculates $P_{pub} = [k]P_1$. Let the master key pair be

$$(mpk, msk) = (P_{pub}, k),$$

*mpk* represents the master public key, and *msk* is the master secret key that must be kept secret.

KeyGen($pp, ID, msk$): Taking as input the system parameter $pp$ and an identifier $ID$. The KGC generates the private key $d_{ID}$ for user $ID$. On the finite field $\mathbb{F}_N$, it first calculates $t_1 = H_1(ID||hid_1, N) + k$. If $t_1 = 0$, *msk* is regenerated. Otherwise, continue to calculate $t_2 = k \cdot t_1^{-1}$. The private key can be calculated from $d_{ID} = [t_2]P_2$.

Enc($pp, ID, M$): Taking as input the system parameter $pp$, an identifier $ID$, and a message $M$. The ciphertext $CT = \{C_1, C_2, C_3\}$ is generated as follows

1.  At first, calculate $Q = [H_1(ID||hid, N)]P_1 + P_{pub}$.
2.  Randomly choose $r \in_R [1, N - 1]$. Calculate

$$C_1 = [r]Q = [r \cdot t_1]P_1,$$

3.  Calculate $g = e(P_{pub}, P_2)$, $w = g^r$, $K = KDF(C_1||w||ID, klen)$. If $K$ is an all-zero bit string, return to the second step; otherwise, The first 128 bits of $K$ are denoted as $K_1$, and the last 128 bits are denoted as $K_2$. The message is encrypted and decrypted by the DEM algorithm, denoted as DEM.Enc and DEM.Dec, respectively.

$$C_2 = \text{DEM.Enc}(M, K_1),$$
$$C_3 = MAC(C_2, K_2).$$

Dec($CT, d_{ID}$): A user with the identifier $ID$, upon receiving a ciphertext $CT = \{C_1, C_2, C_3\}$, performs the following calculations:

1.  Verify if $C_1 \in \mathbb{G}_1$. If the result is false, it outputs $\bot$ and aborts;
2.  Calculate the element $w' = e(C_1, d_{ID})$ in the group $\mathbb{G}_T$;
3.  Calculate $K' = KDF(C_1||w'||ID, klen)$, where $K'$ has its first 128 bits as $K_1'$ and the last 128 bits as $K_2'$;
4.  Calculate $M' = \text{DEM.Dec}(C_2, K_1')$, $C_3' = MAC(C_2, K_2')$. If $C_3' = C_3$, output $M'$.

Bentahar et al. [44] extended the hybrid encryption formalized by Cramer and Shoup [45] to identity-based cryptosystems. Their work showed that an IND-ID-CCA (Indistinguishable against adaptive identity adaptive chosen-ciphertext-attacks) secure IBE can be constructed by an IND-ID-CCA identity-based KEM and a secure DEM.

To maintain consistency in algorithmic style, we adopt the SM4 symmetric encryption algorithm [46] as the DEM algorithm. Similar to the approach taken in the security proofs for SM9-KEM, we do not consider the security of the DEM and other auxiliary functions in SM9. The focus of the security proof primarily lies in the security of the KEM algorithm.

## 4. System Model, Definitions and Security Models

We give the system model of our SM9-Identity-based Encryption with Designated-Position Fuzzy Equality Test scheme, then the formal definition and security models of it.

### 4.1. System Model of IBE-DFET

The system model of our IBE-DFET scheme is illustrated in Figure 1. There are four types of entities in our work, as follows:

1.  Key Generation Center (KGC): Key Generation Center (KGC): This entity is responsible for setting up the system, safeguarding the master secret key, and issuing private keys to users based on their IDs.
2.  User: This entity, as the data owner, can upload the ciphertexts to the cloud server or download ciphertexts for decryption, and grant authorization to testers for designated-position fuzzy equality test.
3.  Cloud server: This entity stores the ciphertexts generated by message senders, allows message receivers to download ciphertexts, and often serves as the tester.

4. Tester: After being authorized, this entity can choose a wildcard set $J$, and conduct an equality test on a ciphertext and a given message.

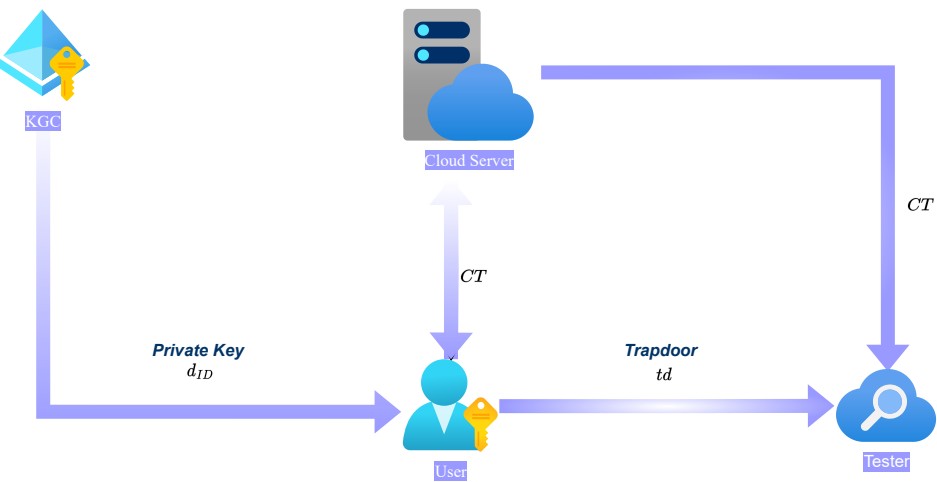

**Figure 1.** System Model of IBE-DFET.

Figure 2 presents a significant feature of our IBE-DFET scheme. A tester can choose a wildcard set and perform fuzzy equality test between the received ciphertext and a given plaintext. When the plaintext corresponding to the ciphertext is completely equal to the existing plaintext at positions outside of the wildcard set, the equality test algorithm outputs 1. If there is at least one different bit outside the wildcard set, the equality test algorithm outputs 0.

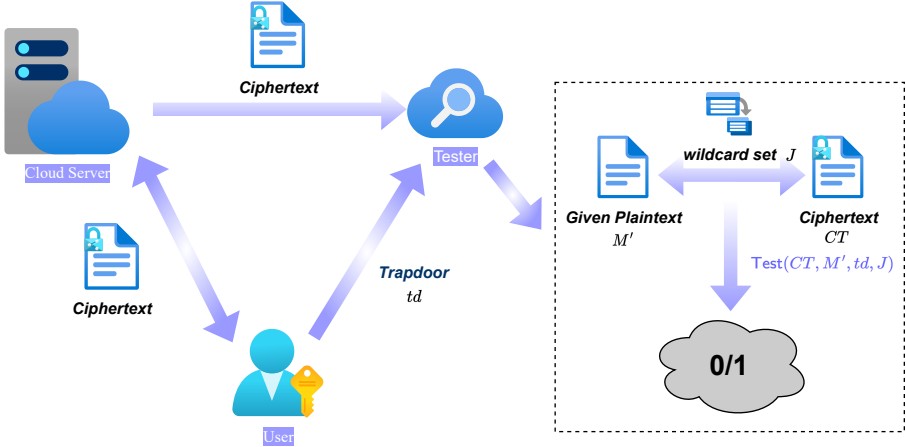

**Figure 2.** Equality Test of CLE-MET-PA.

*4.2. Identity-Based Encryption with Designated-Position Fuzzy Equality Test*

An IBE-DFET system consists of six algorithms as follows:

- $\mathsf{Setup}(1^\lambda)$: This algorithm takes the security parameter $\lambda$ as input and outputs the system parameter $pp$ and the master key pair $(mpk, msk)$.
- $\mathsf{KeyGen}(pp, ID, msk)$: Taking as input the system parameter $pp$ and an identifier $ID$. The KGC generates the private key $d_{ID}$ for user $ID$.
- $\mathsf{Enc}(pp, ID, M)$: Taking as input the system parameter $pp$, an identifier $ID$, and a message $M$. The algorithm generates the ciphertext $CT$.
- $\mathsf{Dec}(CT, d_{ID})$: This algorithm outputs the message $M$ or $\perp$.
- $\mathsf{Aut}(d_{ID})$: This algorithm outputs a token $td$ that authorizes the tester to perform an equality test on the ciphertexts of users who own $d_{ID}$.

- Test($CT, td, M', J$): Taking as input a ciphertext $CT$, a trapdoor $ID$, a sample message $M'$, and a wildcard set $J$. The algorithm checks if $|J| \leq |L|$. If not, it outputs $\bot$ and aborts. Otherwise, it outputs 1, implying that the underlying message of $CT$ is fuzzy equal to $M'$ under the wildcard set $J$, or 0, implying that the messages are not fuzzy equal.

*Correctness:* We can say an IBE-DFET scheme is correct if the following conditions hold.

(1) For any security parameter $\lambda$, and any message $M \in \mathcal{M}$, we have

$$\Pr\left[\mathsf{Dec}(CT, d_{ID}) = M \,\middle|\, \begin{array}{l} pp \leftarrow \mathsf{Setup}(1^\lambda) \\ d_{ID} \leftarrow \mathsf{KeyGen}(pp, ID, msk) \\ CT \leftarrow \mathsf{Enc}(pp, ID, M) \end{array}\right] = 1.$$

(2) For any security parameter $\lambda$, any message $M \in \mathcal{M}$, any wildcard set $J$ satisfying $|J| \leq |L|$, and any message $M'$ satisfying $M \backslash J = M' \backslash J$, we have

$$\Pr\left[\mathsf{Test}(CT, td, M', J) = 1 \,\middle|\, \begin{array}{l} pp \leftarrow \mathsf{Setup}(1^\lambda) \\ d_{ID} \leftarrow \mathsf{KeyGen}(pp, ID, msk) \\ CT \leftarrow \mathsf{Enc}(pp, ID, M) \\ td \leftarrow \mathsf{Aut}(d_{ID}) \end{array}\right]$$

is overwhelming.

### 4.3. Security Models of IBE-DFET

We consider two types of adversaries in IBE-DFET.

- Type-I Adversary: This type of adversary can make trapdoor queries on any user, which means it can test the equality between any ciphertext and a given plaintext. Therefore, we define the goal of a Type-I adversary as recovering the underlying message from the given ciphertext. It is worth noting that it is not necessary to recover the complete message here. Only the positions not contained in the wildcard set need to be recovered.
- Type-II Adversary: This type of adversary can make trapdoor queries on any user except the target user. Therefore, we define the goal of Type-II adversary as distinguishing the underlying message from the given ciphertext with two known messages.

We define two games for these two types of adversaries.

---

**Game 1: F-OW-ID-CCA Game**

---

$$pp \leftarrow \mathsf{Setup}(1^\lambda);$$
$$d_{ID_i} \leftarrow \mathsf{KeyGen}(pp, ID_i, msk) \text{ for } 1 \leq i \leq N;$$
$$M_i/\bot \leftarrow \mathsf{Dec}(CT_i, d_{ID_i});$$
$$td_i \leftarrow \mathsf{Aut}(d_{ID_i});$$
$$ID^* \leftarrow \mathcal{A}^{\mathcal{O}^{\mathsf{KeyGen}}(\cdot), \mathcal{O}^{\mathsf{Dec}}(\cdot), \mathcal{O}^{\mathsf{token}}(\cdot)}(\{ID_i\}_{i=1}^N);$$
$$CT^* \leftarrow \mathsf{Enc}(pp, ID^*, M^*) \text{ for random } M;$$
$$M' \leftarrow \mathcal{A}^{\mathcal{O}^{\mathsf{KeyGen}}(\cdot), \mathcal{O}^{\mathsf{Dec}}(\cdot), \mathcal{O}^{\mathsf{token}}(\cdot)}(\{ID_i\}_{i=1}^N).$$

---

In Game 1, $\mathcal{O}^{\mathsf{KeyGen}}(\cdot)$, $\mathcal{O}^{\mathsf{Dec}}(\cdot)$, $\mathcal{O}^{\mathsf{token}}(\cdot)$ denote the private key oracle, the decryption oracle, and the token oracle, respectively. The adversary is not allowed to make a private-key query on $ID^*$. We define the advantage of the adversary in winning this game as

$$\mathsf{Adv}_{\mathsf{SM9\text{-}IBE\text{-}DFET}}^{\mathsf{F\text{-}OW\text{-}ID\text{-}CCA, Type\text{-}I}}(\lambda) = \Pr[M' \backslash J' = M^* \backslash J'].$$

| Game 2: IND-ID-CCA Game |
|---|

$$pp \leftarrow \mathsf{Setup}(1^\lambda);$$
$$d_{ID_i} \leftarrow \mathsf{KeyGen}(pp, ID_i, msk) \text{ for } 1 \le i \le N;$$
$$M_i / \perp \leftarrow \mathsf{Dec}(CT_i, d_{ID_i});$$
$$td_i \leftarrow \mathsf{Aut}(d_{ID_i});$$
$$(ID^*, M_0^*, M_1^*) \leftarrow \mathcal{A}^{\mathcal{O}^{\mathsf{KeyGen}}(\cdot), \mathcal{O}^{\mathsf{Dec}}(\cdot), \mathcal{O}^{\mathsf{token}}(\cdot)}(\{ID_i\}_{i=1}^N);$$
$$CT^* \leftarrow \mathsf{Enc}(pp, ID^*, M_b^*) \text{ for } b \in \{0, 1\};$$
$$b' \leftarrow \mathcal{A}^{\mathcal{O}^{\mathsf{KeyGen}}(\cdot), \mathcal{O}^{\mathsf{Dec}}(\cdot), \mathcal{O}^{\mathsf{token}}(\cdot)}(\{ID_i\}_{i=1}^N).$$

In Game 2, the adversary is restricted to make a private-key query or a token query on $ID^*$. We define the advantage of the adversary in winning this game as

$$\mathsf{Adv}_{\mathsf{SM9\text{-}IBE\text{-}DFET}}^{\mathsf{IND\text{-}ID\text{-}CCA,Type\text{-}II}}(\lambda) = \Pr[b' = b] - 1/2.$$

## 5. The Proposed SM9-IBE-DFET Scheme

In identity-based encryption with designated-position fuzzy equality test (IBE-DFET), a tester can choose a wildcard set *J*. After being authorized by a user, the tester is enabled to perform a fuzzy equality test between the ciphertext of the user and a given message, while the positions in the wildcard set do not affect the result of the equality test.

### 5.1. Our Construction

Setup$(1^\lambda)$: Taking as input a security parameter $\lambda$, the setup algorithm generates the public parameter

$$pp = \{\mathbb{G}_1, \mathbb{G}_2, \mathbb{G}_T, P_1, P_2, e, hid_1, hid_2, H_1, H_2, H_3\},$$

where $\mathbb{G}_1$ and $\mathbb{G}_2$ are two additive cyclic groups of order *N*, and $P_1 \in \mathbb{G}_1$ and $P_2 \in \mathbb{G}_2$ are the generators of the two groups. $\mathbb{G}_T$ is a multiplicative group with order *N*. *e* represents the bilinear pairing: $\mathbb{G}_1 \times \mathbb{G}_2 \to \mathbb{G}_T$. $hid_1$ and $hid_2$ are two distinct identifiers for private key generation functions. $H_1$: $\{0,1\}^* \to \mathbb{Z}_N^*$, $H_2$: $\{0,1\}^* \to \mathbb{G}_1$, $H_3$: $\mathbb{G}_1 \to \mathbb{G}_T$ are cryptographic hash functions. Afterward, the Key Generation Center (KGC) randomly selects $k \in [1, N-1]$, and calculates $P_{pub} = [k]P_1$. Let the master key pair be:

$$(mpk, msk) = (P_{pub}, k),$$

*mpk* represents the master public key, and *msk* is the master secret key that must be kept secret. Additionally, within the SM9 framework, there are several auxiliary functions, along with the DEM algorithm SM4 that we selected. Specifically: SM4.Enc and SM4.Dec represent the standard SM4 algorithm for encryption and decryption, respectively. MAC denotes the message authentication code function. The key derivation function KDF: $\{0,1\}^* \to klen$, where $klen = 256$.

KeyGen$(pp, ID, msk)$: Taking as input the system parameter *pp* and an identifier *ID*. The KGC generates the private key $d_{ID}$ for user *ID*. On the finite field $\mathbb{F}_N$, it first calculate $t_1 = H_1(ID||hid_1, N) + k$, $t_3 = H_1(ID||hid_2, N) + k$. If $t_1 = 0$ or $t_3 = 0$, *msk* is regenerated. Otherwise, continue to calculate $t_2 = k \cdot t_1^{-1}$, $t_4 = k \cdot t_3^{-1}$. The private key can be calculated from $d_1 = [t_2]P_2$, $d_2 = [t_4]P_2$. The complete private key pair of the user is

$$d_{ID} = (d_1, d_2) = ([t_2]P_2, [t_4]P_2).$$

Enc$(pp, ID, M)$: Taking as input the system parameter *pp*, an identifier *ID*, and a message *M*. The message can be represented in the bit form: $M = \{M_1, M_2, \cdots, M_n\}$.

The ciphertext is generated as follows:

1.  At first, calculate $Q_1 = [H_1(ID||hid_1, N)]P_1 + P_{pub}$, $Q_2 = [H_1(ID||hid_2, N)]P_1 + P_{pub}$;

2. Randomly choose $r_1, r_2 \in_R [1, N-1]$. Calculate

$$C_1 = [r_1]Q_1 = [r_1 \cdot t_1]P_1, C_4 = [r_2]Q_2 = [r_2 \cdot t_3]P_1,$$

3. Calculate $g = e(P_{pub}, P_2)$, $w_1 = g^{r_1}$, $K = KDF(C_1||w_1||ID, klen)$, if $K$ is an all-zero bit string, return to the second step; otherwise, The first 128 bits of $K$ are denoted as $K_1$, and the last 128 bits are denoted as $K_2$. Calculate

$$C_2 = \text{SM4.Enc}(M, K_1),$$

$$C_3 = MAC(C_2, K_2);$$

4. Calculate

$$\left\{ C_{5,l} = \sum_{i=1}^{n} i^l H_2(M_i||i) \cdot r_2 Q_2 \right\}_{l=\{0,1,\cdots,L\}}$$

5. Calculate $w_2 = g^{r_2}$, $C_6 = H_3(C_1||C_2||C_3||C_4||\{C_{5,l}\}_{l=0}^{L}||w_2)$.

   The complete ciphertext $CT$ is

$$CT = \{C_1, C_2, C_3, C_4, \{C_{5,l}\}_{l=0}^{L}, C_6\}.$$

Dec($CT, d_{ID}$): A user with the identifier $ID$, upon receiving a ciphertext $CT = \{C_1, C_2, C_3, C_4, \{C_{5,l}\}_{l=0}^{L}, C_6\}$, performs the following calculations:

1. Verify if $C_1 \in \mathbb{G}_1$. If the result is false, it outputs $\perp$ and aborts;
2. Calculate the element $w_1' = e(C_1, d_1)$, $w_2' = e(C_4, d_2)$ in the group $\mathbb{G}_T$. Check if

$$C_6 = H_3(C_1||C_2||C_3||C_4||\{C_{5,l}\}_{\{l=0,1,\dots,L\}}||w_2'),$$

   if not, it outputs $\perp$ and aborts;
3. Calculate $K' = KDF(C_1||w_1'||ID, klen)$, where $K'$ has its first 128 bits as $K_1'$ and the last 128 bits as $K_2'$;
4. Calculate $M' = \text{SM4.Dec}(C_2, K_1')$, $C_3' = MAC(C_2, K_2')$. If $C_3' = C_3$, output $M'$.

Aut($d_{ID}$): The trapdoor sent by the user to the tester is given by:

$$td = d_2 = [t_4]P_2.$$

Test($CT, td, M', J$): Upon receiving the ciphertext $CT = \{C_1, C_2, C_3, C_4, \{C_{5,l}\}_{\{l=0,1,\dots,L\}}, C6\}$ from the authorized user, along with the corresponding trapdoor $td = d_2$, the plaintext $M'$ for comparison, and the wildcard set $J = \{j_1, j_2, \cdots, j_m\} \subsetneq \{1, 2, \cdots, n\}$, the tester performs the following calculations:

1. Verify if $C_4 \in \mathbb{G}_1$. If the result is false, it outputs $\perp$ and aborts;
2. Calculate the element $w_2' = e(C_4, td)$ in the group $\mathbb{G}_T$. If

$$C_6 = H_3(C_1||C_2||C_3||C_4||\{C_{5,l}\}_{\{l=0,1,\dots,L\}}||w_2'),$$

   then $w_2 = w_2'$;
3. For each $0 \leq l \leq m$, calculate

$$a_{m-l} = (-1)^l \sum_{1 \leq i_1 \leq i_2 \cdots \leq i_l \leq m} j_{i_1} j_{i_2} \cdots j_{i_l};$$

4. Calculate

$$X = e\left( \sum_{l=0}^{m} a_l(C_{5,l}), td \right),$$

and

$$Y = (w_2)^{\sum\limits_{i=1}^{n} H_2(M_i'\|i) \sum\limits_{l=0}^{m} a_l i^l},$$

where $M_i'$ is a bit of the given plaintext $M'$ which is $(M_1', \ldots, M_n')$ in bit form. If $X = Y$, then $M\backslash J = M'\backslash J$, and the algorithm outputs 1; otherwise, if $X \neq Y$, then $M\backslash J \neq M'\backslash J$, and the algorithm outputs 0.

*5.2. Correctness of SM9-IBE-DFET*

We analyze the correctness of the proposed SM9-IBE-DFET construction as below.

(1) In the decryption algorithm, denoted as Dec, the decryption process computes the following: For any legitimate ciphertext $CT = \{C_1, C_2, C_3, C_4, \{C_{5,l}\}_{l=0}^{L}, C_6\}$, calculate the element $w_1'$ on group $\mathbb{G}_T$:

$$\begin{aligned}
w_1' &= e(C_1, d_1) \\
&= e([r_1 \cdot t_1]P_1, [t_2]P_2) \\
&= e(k \cdot P_1, P_2)^{r_1 \cdot t_2^{-1} t_2} \\
&= e(P_{pub}, P_2)^{r_1}
\end{aligned}$$

Compute $K' = KDF(C_1\|w_1'\|ID, klen)$, where $K'$ has its first 128 bits as $K_1'$ and the last 128 bits as $K_2'$. Clearly, if $w_1' = w_1$, then $K' = K$. Additionally, $C_3' = MAC(C_2, K_2')$, and if $w_1' \neq w_1$, it is challenging to obtain $C_3' = C_3$. Thus, we can verify the correctness of decryption, implying that

$$\Pr\left[\mathsf{Dec}(CT, d_{ID}) = M\right] = 1.$$

(2) In the test algorithm, denoted as Test, for any legal ciphertext $CT = \{C_1, C_2, C_3, C_4, \{C_{5,l}\}_{l=0}^{L}, C_6\}$, the corresponding user trapdoor is $td = d_2$, plaintext $M'$ used for comparison, and wildcard set $J = \{j_1, j_2, \cdots, j_m\} \subsetneq \{1, 2, \cdots, n\}$.

Calculate

$$\begin{aligned}
X &= e\left(\sum_{l=0}^{m} a_l(C_{5,l}), td\right) \\
&= e\left(\sum_{l=0}^{m} a_l \sum_{i=1}^{n} i^l H_2(M_i\|i) \cdot r_2 Q_2, [t_4]P_2\right) \\
&= e([k]P_1, P_2)^{\sum\limits_{l=0}^{m} a_l \sum\limits_{i=1}^{n} i^l H_2(M_i\|i) \cdot r_2 t_4^{-1} t_4} \\
&= e(P_{pub}, P_2)^{r_2 \sum\limits_{i=1}^{n} H_2(M_i\|i) \prod\limits_{i\in J} (i-j)}
\end{aligned}$$

Then calculate

$$\begin{aligned}
Y &= (w_2)^{\sum\limits_{i=1}^{n} H_2(M_i'\|i) \sum\limits_{l=0}^{m} a_l i^l} \\
&= e(P_{pub}, P_2)^{r_2 \sum\limits_{i=1}^{n} H_2(M'_i\|i) \prod\limits_{i\in J} (i-j)}
\end{aligned}$$

From this, it can be seen that if $M\backslash J = M'\backslash J$, then $X = Y$, and the algorithm outputs 1. We have

$$\mathsf{Test}(CT, td, M', J) = 1$$

with overwhelming probability.

Otherwise, if $X \neq Y$, then $M\backslash J \neq M'\backslash J$, and the algorithm outputs 0. We have

$$\mathsf{Test}(CT, td, M', J) = 0$$

with overwhelming probability.

## 6. Security Proof

We employ the proof technique introduced in [39,41] to perform the security proof of our SM9-IBE-DFET scheme.

**Theorem 1.** *For any PPT Type-I adversary, our SM9-IBE-DFET scheme is F-OW-ID-CCA (fuzzy-one-way against adaptive identity adaptive chosen-ciphertext-attacks) secure based on the soundness of $Gap - \tau - BCAA1_{1,2}$ assumption in the random oracle model.*

**Proof of Theorem 1.** We divide the security proof of **Theorem 1** into two lemmas. In **Lemma 1**, we prove the OW-ID-CCA security of our SM9-IBE-DFET scheme. In **Lemma 2**, we prove our OW-ID-CCA security can be reduced to F-OW-ID-CCA security.  □

**Lemma 1.** *Our SM9-IBE-DFET construction is provably secure in the OW-ID-CCA security model against Type-I adversary if functions $H_1$, $H_2$, $H_3$ and KDF are random oracles.*

**Proof.** Assume there is an adversary $\mathcal{A}_1$ who can break the OW-ID-CCA security of our SM9-IBE-DFET scheme with advantage $\epsilon_1(k)$, we can construct a simulator $\mathcal{B}$ to break the $Gap - \tau - BCAA1_{1,2}$ problem with non-negligible advantage.

Given an instance of the $Gap - q_1 - BCAA1_{1,2}$ problem: $\left(P_1, P_2, [k]P_1, h_0, (h_{1,1}, [\frac{k}{h_{1,1}+k}]),\right.$
$\left.\dots, (h_{\frac{q_1}{2},1}, [\frac{k}{h_{\frac{q_1}{2},1}+k}]), (h_{1,2}, [\frac{k}{h_{1,2}+k}]), \dots, (h_{\frac{q_1}{2},2}, [\frac{k}{h_{\frac{q_1}{2},2}+k}])\right)$ where $h_{i,j} \in_R \mathbb{Z}_N^*$ for $0 \le i \le \frac{q_1}{2}, j \in \{1,2\}$, and $\mathcal{O}^{\mathsf{DBIDH}}$ is the $DBIDH_{1,1}$ oracle. $\mathcal{B}$ runs $\mathsf{Setup}(1^\lambda)$ to generate the public parameter $pp = \{\mathbb{G}_1, \mathbb{G}_2, \mathbb{G}_T, P_1, P_2, e, hid_1, hid_2, H_1, H_2, H_3\}$, $P_{pub} = [k]P_1$, $g = e(P_{pub}, P_2)$. $\mathcal{B}$ randomly chooses $1 \le I \le q_1 + 1$ and interacts with $\mathcal{A}_1$ as follows:

- $\mathcal{O}^{H_1}$: $\mathcal{B}$ maintains two lists $\mathcal{L}_{H_{1,j}}$ for $j \in \{1,2\}$ of tuples $(ID_i, h_{i,j}, d_{i,j})$ as explained below. When $\mathcal{A}_1$ queries $\mathcal{O}^{H_1}$ on $(ID_i, hid_j)$. $\mathcal{B}$ responds as follows:

  - If $ID_i$ is on $\mathcal{L}_{H_{1,j}}$ with a tuple $(ID_i, h_{i,j}, d_{i,j})$, $\mathcal{B}$ returns with $H_1(ID_i, hid_j) = h_{i,j}$.
  - Otherwise, if the query is on the $I$-th distinct $ID$, then $\mathcal{B}$ stores $(ID_I, h_0, \perp)$ into the list $\mathcal{L}_{H_{1,j}}$ and responds with $H_1(ID_I, hid_j) = h_0$.
  - Otherwise, $\mathcal{B}$ selects a random integer $h_{i,j}$ which was not chosen before from the given $Gap - q_1 - BCAA1_{1,2}$ instance, stores $(ID_i, h_{i,j}, d_{i,j})$ into $\mathcal{L}_{H_{1,j}}$ and responds with $H_1(ID_i, hid_j) = h_{i,j}$.

- $\mathcal{O}^{H_2}$: $\mathcal{B}$ maintains a list $\mathcal{L}_{H_2}$ of $(U_i, \sigma_i)$. When $\mathcal{A}_1$ queries $\mathcal{O}^{H_2}$ on $U_i$. $\mathcal{B}$ responds as follows:

  - If $Q_i$ is on $\mathcal{L}_{H_2}$ with a tuple $(U_i, \sigma_i)$, $\mathcal{B}$ returns with $H_2(U_i) = \sigma_i$.
  - Otherwise, $\mathcal{B}$ selects a random integer $\sigma_i \in \mathbb{Z}_N^*$, stores $(U_i, \sigma_i)$ into $\mathcal{L}_{H_2}$ and responds with $H_2(U_i) = \sigma_i$.

- $\mathcal{O}^{H_3}$: $\mathcal{B}$ maintains a list $\mathcal{L}_{H_3}$ of $(V_i, \eta_i)$. When $\mathcal{A}_1$ queries $\mathcal{O}^{H_3}$ on $V_i$. $\mathcal{B}$ responds as follows:

  - If $V_i$ is on $\mathcal{L}_{H_3}$ with a tuple $(V_i, \eta_i)$, $\mathcal{B}$ returns with $H_3(V_i) = \eta_i$.
  - Otherwise, $\mathcal{B}$ selects a random bitstring $\eta \in \{0,1\}^\lambda$, stores $(V_i, \eta_i)$ into $\mathcal{L}_{H_3}$ and responds with $H_3(V_i) = \eta_i$.

- $\mathcal{O}^{\mathsf{KDF}}$: $\mathcal{B}$ maintains a list $\mathcal{L}_{KDF}$ of tuples $(\langle ID_i, hid_j, W_i, C_i \rangle, K_{i,j})$. $\mathcal{B}$ interacts with $\mathcal{A}_1$ on a query of $(ID_i, hid_j, W_i, C_i)$ as follows:

  - If $(\langle ID_i, hid_j, W_i, C_i \rangle, K_{i,j})$ is on $\mathcal{L}_{KDF}$, $\mathcal{B}$ returns with $KDF(ID_i, hid_j, W_i, C_i) = K_{i,j}$.
  - Otherwise, $\mathcal{B}$ searches $\mathcal{L}_{H_{1,j}}$ with entry $(ID_i, hid_j)$, if $(ID_i, hid_j)$ is not on the list, $\mathcal{B}$ makes a query of $(ID_i, hid_j)$ on $\mathcal{O}^{H_1}$.

    * If $d_{i,j} = \perp$, $\mathcal{B}$ makes a query on $\mathcal{O}^{\mathsf{DBIDH}}$ with $([k]P_1, P_2, [h_0 + k]P_1, C_i, X_i)$.
      · If $\mathcal{O}^{\mathsf{DBIDH}}$ returns 1, and a tuple indexed by $(ID_i, hid_j, C_i)$ is on list $\mathcal{L}_D$, $\mathcal{B}$ returns $K_{i,j}$ after storing $(\langle ID_i, hid_j, W_i, C_i \rangle, K_{i,j})$ into $\mathcal{L}_{KDF}$.

·      Otherwise, $\mathcal{B}$ randomly chooses a bitstring $K_{i,j} \in \{0,1\}^{256}$ and adds $(\langle ID_i, hid_j, W_i, C_i \rangle, K_{i,j})$ into $\mathcal{L}_{KDF}$. Then returns $K_{i,j}$ to $\mathcal{A}_1$.

∗      Otherwise, $\mathcal{B}$ randomly chooses a bitstring $K_{i,j} \in \{0,1\}^{256}$ and adds $(\langle ID_i, hid_j, W_i, C_i \rangle, K_{i,j})$ into $\mathcal{L}_{KDF}$. Then returns $K_{i,j}$ to $\mathcal{A}_1$.

-    $\mathcal{O}^{\mathsf{KeyGen}}$: $\mathcal{B}$ searches $\mathcal{L}_{H_{1,1}}$ for entry $(ID_i, hid_1)$. If it is not in the list, $\mathcal{B}$ makes a query on $\mathcal{O}^{H_1}$ with $(ID_i, hid_1)$. If $d_{i,1} \neq \bot$, $\mathcal{B}$ adds $(ID_i, hid_1, h_{i,1}, d_{i,1})$ into list $\mathcal{L}_{H_{1,1}}$ and return $d_{i,1}$ to $\mathcal{A}_1$. If $d_{i,1} = \bot$. $\mathcal{B}$ aborts the game. (Event $E_1$)

-    $\mathcal{O}^{\mathsf{Token}}$: $\mathcal{B}$ searches $\mathcal{L}_{H_{1,2}}$ for entry $(ID_i, hid_2)$. If it is not in the list, $\mathcal{B}$ makes a query on $\mathcal{O}^{H_1}$ with $(ID_i, hid_2)$. If $d_{i,2} \neq \bot$, $\mathcal{B}$ adds $(ID_i, hid_2, h_{i,2}, d_{i,2})$ into list $\mathcal{L}_{H_{1,2}}$ and return $d_{i,2}$ to $\mathcal{A}_1$. If $d_{i,2} = \bot$. $\mathcal{B}$ aborts the game. (Event $E_2$)

-    $\mathcal{O}^{\mathsf{Dec}}$: $\mathcal{B}$ maintains a list $\mathcal{L}_{Dec}$ of entries in form $(ID_i, hid_j, C_i, K_{i,j})$. On a query $(ID_i, hid_j, C_i)$. $\mathcal{B}$ searches $\mathcal{L}_{H_{1,j}}$ for the entry indexed by $(ID_i, hid_j)$. If it is not in the list, $\mathcal{B}$ makes a query on $\mathcal{O}^{H_1}$ with $(ID_i, hid_j)$. Then, $\mathcal{B}$ responds depending on the value $d_{i,j}$.

     ▪    If $d_{i,j} \neq \bot$, $\mathcal{B}$ computes $g^r = e(C_i, d_{i,j})$, makes a query of $(ID_i, hid_j, g^r, C_i)$ on $\mathcal{O}^{\mathsf{KDF}}$. Then $\mathcal{B}$ returns $K_{i,j}$ to $\mathcal{A}_1$.

     ▪    Otherwise ($di, j = \bot$),

         ∗    If there is a tuple indexed by $(ID_i, hid_j, C_i)$ is on $\mathcal{L}_{Dec}$, return $K_{i,j}$.

         ∗    Otherwise, $\mathcal{B}$ randomly chooses $K_{i,j} \in \{0,1\}^{256}$ and stores $(ID_i, hid_j, C_i, K_{i,j})$ into $\mathcal{L}_{Dec}$.

**Challenge:** At some point, $\mathcal{A}_1$ will return a challenge identifier $ID^*$. $\mathcal{B}$ searches $\mathcal{L}_{H_{1,j}}$ for the items $(ID_i, hid_1)$ and $(ID_i, hid_2)$. If both $d_{i,1}$ and $d_{i,2}$ are not equal to $\bot$, $\mathcal{B}$ aborts (Event $E_3$). $\mathcal{B}$ chooses a random value $y \in \mathbb{Z}_N^*$ and a random bistring $K^* = \{0,1\}^{256}$, returns $(K^*, [y]P_1)$ as the challenge.

**Guess:** Once $\mathcal{A}_1$ outputs its guess, $\mathcal{B}$ answers the $Gap - q_1 - BCAA1_{1,2}$ challenge in the following way.

•   For the tuple $(\langle ID^*, hid_j, W_i, [y]P_1 \rangle, K_{i,j})$ in list $\mathcal{L}_{KDF}$, $\mathcal{B}$ makes queries on $\mathcal{O}^{\mathsf{DBIDH}}$ with $([k]P_1, P_2, [h_0 + k]P_1, [y]P_1), W_i$, if $\mathcal{O}^{\mathsf{DBIDH}}$ returns 1, $\mathcal{B}$ outputs $W_i^{\frac{1}{y}}$ as the answer to the $Gap - q_1 - BCAA1_{1,2}$ challenge.

•   If there is no such tuple in $\mathcal{L}_{KDF}$. $\mathcal{B}$ aborts. (Event $E_4$)

**Analysis:** As long as $\mathcal{B}$ does not abort, from the perspective of $\mathcal{A}_1$, $\mathcal{B}$'s responses to $\mathcal{A}$'s queries on $H_1$, $H_2$, $H_3$, and $KDF$ are all uniform and independent, indistinguishable from a real attack. Now we evaluate the probability that $\mathcal{B}$ does not abort the game. Event $E_4$ implies $e(C^*, [\frac{k}{h_0+k}]P_2)$ is not queried on $\mathcal{O}^{\mathsf{KDF}}$. Obviously, we have

$$\Pr[\mathcal{A}_1 \ wins] = \Pr[\mathcal{A}_1 \ wins | E_4] \Pr[E_4] + \Pr[\mathcal{A}_1 \ wins | \overline{E_4}] \Pr[\overline{E_4}]$$

$$\leq \frac{1}{2}(1 - \Pr[\overline{E_4}]) + \Pr[\overline{E_4}]$$

$$= \frac{1}{2} + \frac{1}{2} \Pr[\overline{E_4}].$$

$$\Pr[\mathcal{A}_1 \ wins] \geq \Pr[\mathcal{A}_1 \ wins | E_4] \Pr[E_4]$$

$$= \frac{1}{2}(1 - \Pr[\overline{E_4}])$$

$$= \frac{1}{2} - \frac{1}{2} \Pr[\overline{E_4}].$$

Therefore, $\Pr[\overline{E_4}] \geq \epsilon_1(k)$. Due to the game rules, $\overline{E_3}$ implies $\overline{E_2}$ and $\overline{E_1}$. Overall, we have

$$\Pr[\mathcal{B} \ wins] = \Pr[\overline{E_1} \wedge \overline{E_2} \wedge \overline{E_3}] \geq \frac{\epsilon_1(k)}{q_1 + 1}.$$

This completes the security analysis of our SM9-IBE-DFET scheme. □

**Lemma 2.** *If our proposed SM9-IBE-DFET is OW-ID-CCA secure, it is then F-OW-ID-CCA secure.*

**Proof.** Suppose there is an adversary $\mathcal{A}_1$ that can break the F-OW-ID-CCA security of our SM9-IBE-DFET scheme with non-negligible advantage $\epsilon_2(k)$, we can construct a simulator $\mathcal{B}$ to break the OW-ID-CCA security running $\mathcal{A}_1$ as a subroutine.

The interaction between $\mathcal{B}$ and $\mathcal{A}_1$ is the same as the interaction process in the proof of **Lemma 1**, with the addition of the following steps.

**Attack:** When $\mathcal{A}_1$ outputs a message $M' \in \{0,1\}^n$ and a wildcard set $J' = \{j'_1, \ldots, j'_m\} \subsetneqq N$, where $N = \{1, 2, \cdots, n\}$. and $|J'| \leq L^* = U$.

**Solution:** $\mathcal{B}$ transforms $M'$ into bit form $M' = M'_1, \ldots, M'_n$, picks random bits $\{s_1, \ldots, s_m\} \in \{0,1\}$ and resets

$$M_{j'_i} = s_i \text{ for each } i \in \{1, \ldots, m\}.$$

It then sends the new message to $\mathcal{A}_1$ as the underlying message of the challenge ciphertext.

**Analysis:** We have $M' \backslash J' = M^* \backslash J'$ will hold with non-negligible advantage $\epsilon_2(k)$ since $\mathcal{A}_1$ has non-negligible advantage $\epsilon_2(k)$ in breaking the F-OW-ID-CCA security of our scheme. We can guess all the positions in $J'$ with the probability $\frac{1}{2^m}$. Therefore, the probability of breaking the OW-ID-CCA security is $\frac{\epsilon_2(k)}{2^m} \geq \frac{\epsilon_2(k)}{2^U}$. Since $2^U$ is polynomial size, $\mathcal{B}$ can break the OW-ID-CCA security with a non-negligible advantage. $\square$

**Theorem 2.** *For any PPT Type-II adversary, our SM9-IBE-DFET scheme is IND-ID-CCA secure based on the $Gap - \tau - BCAA1_{1,2}$ assumption in the random oracle model.*

**Proof of Theorem 2.** Assume there exists an adversary $\mathcal{A}_2$ to attack the IND-ID-CCA security of our scheme.

**Game $G_0$:**

$G_0$ is the original Game 2 defined in Section 4.3, except $H_2, H_3$ are random oracles. And the oracles work as follows:

- $\mathcal{O}^{\mathsf{KeyGen}}$: On inputting an identifier $ID_i \neq ID^*$, calculate the correponding private key $d_{ID_i}$ and return to $\mathcal{A}_2$.

- $\mathcal{O}^{\mathsf{Dec}}$: On inputting a ciphertext $CT_i \neq CT^*$ of $ID_i$, calculate the message $M_i$ with the corresponding private key $d_{ID_i}$ and return to $\mathcal{A}_2$.

- $\mathcal{O}^{\mathsf{Token}}$: On inputting a $ID_i \neq ID^*$, return the $td_i = d_{i,2}$ to $\mathcal{A}_2$.

- $\mathcal{O}^{H_2}$: On inputting a bitstring from $\{0,1\}^*$, a *compatible* random value is returned, which means if the same random value is returned if the input bitstring is same.

- $\mathcal{O}^{H_3}$: On inputting a bitstring from $\{0,1\}^*$, a compatible random value is returned.

**Game $G_1$:**

$G_1$ and $G_0$ are almost the same, except for one oracle $\mathcal{O}^{H_3}$:

- $\mathcal{O}^{H_3}$: Let $T_{H_3} = \varnothing$. On input a hash query $W_{1,3}$, look for the entry in $T_{H_3}$; if it does not exist, return a random value $h_3 \in \{0,1\}^\lambda$, and add $\{W_{1,3}, h_3\}$ into $T_{H_3}$. We have $C_6^* = h_3^*$.

Due the idealness of the random oracle, $G_1$ is identical to $G_0$.

**Game $G_2$:**

$G_2$ and $G_1$ are almost the same, except for one oracle $\mathcal{O}^{H_2}$:

- $\mathcal{O}^{H_2}$: Let $T_{H_2} = \emptyset$. On inputting a hash query $W_{2,2}$, look for the entry in $T_{H_2}$; if ti does not exist, return a random value $h_2 \in \mathbb{Z}_N$, and add $\{W_{2,2}, h_2\}$ into $T_{H_2}$. We have

$$\left\{ C^*_{5,l} = \sum_{i=1}^{n} i^l h^*_{i,2} \cdot r_2 Q_2 \right\}_{l=\{0,1,\cdots,L\}}.$$

Due the idealness of the random oracle, $\mathbf{G_2}$ is identical to $\mathbf{G_1}$.

**Game $\mathbf{G_3}$**

$\mathbf{G_3}$ and $\mathbf{G_2}$ are almost the same, except in the challenge phase: randomly choose a value $R^* \in \mathbb{G}_1$, let $C^*_4 = R^*$. According to the construction in Section 5.1, $C_4 = [r_2]Q_2 = [r_2 \cdot t_3]P_1$. From the perspective of $\mathcal{A}_2$, due to the hardness of the discrete logarithm problem, $R^*$ and $[r^*_2 \cdot t_3]P_1$ are indistinguishable when the random number $r^*_2$ is unknown. Therefore, **Game $\mathbf{G_3}$** is identical to $\mathbf{G_2}$.

After successfully simulating $C^*_4$, $C^*_5$, and $C^*_6$ in the ciphertext $CT^*$, the structure of $C^*_1$, $C^*_2$, and $C^*_3$ is completely identical to that of the SM9-IBE scheme. Previously, Cheng et al. [41] have proven the CCA security of SM9-IBE under the random oracle model. Therefore, our security model (which is **Game $\mathbf{G_0}$**) shares the same level of security as SM9-IBE (which is **Game $\mathbf{G_3}$**): assuming the soundness of $Gap - \tau - BCAA1_{1,2}$ problem and that both $H_1$ and $KDF$ are random oracles, thus obtaining IND-ID-CCA security. Therefore, our SM9-IBE-DFET scheme achieves IND-ID-CCA against any PPT Type-II adversary under the hardness of $Gap - \tau - BCAA1_{1,2}$ assumption in the random oracle model. $\square$

## 7. Performance Analysis of SM9-IBE-DFET

We have visually compared our scheme with several existing ones, including [15,38,39]. Ref. [15] is a standard IBEET scheme that does not support a fuzzy equality test; Ref. [38] is a fuzzy matching scheme, with its fuzzy matching algorithm based on edit distance calculation to achieve a fuzzy equality test. Therefore, it does not support fuzzy matching at designated positions; Ref. [39] is the PKE-DFET scheme, which has a certificate management problem because it is a public key encryption scheme. Additionally, due to the lack of an authorization algorithm, it cannot achieve IND security. The results are presented in Table 1.

**Table 1.** Comparison among several equality test schemes.

| Schemes | [15] | [38] | [39] | Ours |
|---|---|---|---|---|
| Enc | 6E + 3H + 2P | $(3n+5)$E + $(2n+1)$H | $(nL+n+2)$E + $(n+1)$H | $(nL+n+4)$E$^{(+)}$ + 2E + $(n+2)$H + 1P |
| Dec | 4E + 3H + 2P | $2n$E + $2n$P | $(nL+n+2)$E + $(n+1)$H | 2H + 2P |
| Test | 2E + 4P | $3n$E + $(2n+2)$P | $2(m+1)$E + 2P | $(m+1)$E$^{(+)}$ + E + $(n+1)$H + 2P |
| $\lvert CT \rvert$ | $5\lvert\mathbb{G}\rvert + \lvert\mathbb{Z}_p\rvert$ | $(2n+4)\lvert\mathbb{G}\rvert$ | $(L+2)\lvert\mathbb{G}\rvert + 2\lvert\mathbb{Z}_p\rvert + \{0,1\}^n$ | $(L+3)\lvert\mathbb{G}\rvert + \{0,1\}^{2n+\lambda}$ |
| Security | OW-ID-CCA | IND-CPA | F-OW-CCA | F-OW/IND-ID-CCA |
| Fuzzy | $\times$ | Designated-Distance | Designated-Position | Designated-Position |
| AntiCM | $\checkmark$ | $\times$ | $\times$ | $\checkmark$ |
| Aut-type | AoN-type | Ciphertext-level | $\times$ | AoN-type(improved) |

E, E$^{(+)}$, H, and P represent the computation cost of an exponential operation on multiplicative group, an exponential operation on additive group, a hash operation, and a pairing operation. $\lvert\mathbb{Z}_p\rvert$, $\lvert\mathbb{G}\rvert$ represent the bit length of a group element in $\mathbb{Z}_p$, $\mathbb{G}$ respectively. $n$: The size of a message. $L$: Maximum allowable size of the wildcard set. AntiCM represents the anti-certificate management feature. Aut-type represents the authorization type. "$\checkmark$" indicates that the scheme supports the feature, while "$\times$" indicates that the scheme does not possess the feature.

The comparison of these schemes was based on algorithm computational cost of encryption, decryption, and testing. Furthermore, we also consider other metrics like ciphertext size, security level, whether they support fuzzy equality tests, the ability to address the certificate management problem, and the type of authorization. In the computation of algorithm complexity, we primarily considered metrics such as exponentiation calculations, hash calculations, and bilinear pairings, while efficient operations like addition, multiplication, and XOR were not included.

As shown in Table 1, our SM9-IBE-DFET scheme inherits the advantages of traditional IBEET algorithms, effectively addressing the certificate management problem. Moreover, our scheme employs the designated-position fuzzy equality test, granting users the flexibility to adjust the positions requiring fuzzy matching, thereby offering greater freedom. In comparison to other schemes, the inclusion of an authorization step enables us to achieve IND-ID-CCA security when the adversary does not have the trapdoor of the challenge ciphertext. Taking into account the authorization approach, our scheme utilizes an improved AoN-type authorization, which retains the flexibility of AoN-type authorization while also restricting the testers' selection of objects for equality test, thereby enhancing the security of the scheme. In terms of computational complexity, our scheme also inherits the advantages of the SM9 algorithm, particularly in decryption algorithms, giving it a significant edge over other schemes and making it more convenient for deployment in computationally constrained scenarios. In summary, as shown in Table 1, our scheme offers fuzzy matching feature compared to the standard IBEET scheme, along with a more flexible authorization method and higher security. Compared to other fuzzy equality test schemes, our approach combines the advantages of IBE cryptosystems, an outstanding fuzzy matching type, an efficient and secure authorization approach, and the highest level of security.

## 8. Conclusions

In this study, we introduce the concept of identity-based encryption with designated-position fuzzy equality test (IBE-DFET), which integrates the functionality of designated-position fuzzy equality test into the IBE framework. Our proposed scheme is constructed utilizing the SM9-IBE algorithm, forming an SM9-IBE-DFET scheme. Notably, within our work, a tester is restricted to conducting fuzzy equality tests solely between its own plaintext and the ciphertext of a designated user, but not equality tests between the ciphertexts of different users, even if such users have authorized the tester by sharing their respective trapdoors. We formalized the system model and two security models for our SM9-IBE-DFET scheme, and subsequently demonstrates that our scheme is robust, achieving F-OW-ID-CCA/IND-ID-CCA security against adversaries with/without the trapdoor of the challenge ciphertext.

**Author Contributions:** Conceptualization, S.D. and Z.Z.; methodology, S.D. and Z.Z.; writing—original draft preparation, S.D.; writing—review and editing, W.G. and S.Z.; supervision, B.W. All authors have read and agreed to the published version of the manuscript.

**Funding:** This research was funded by the National Key R&D Program of China under Grant No. 2023YFB4403500, the National Natural Science Foundation of China under Grant 61972457, 62102299, 62002288, U19B2021, 62272362, 62202363, and the Youth Innovation Team of Shaanxi Universities, Science and Technology on Communication Security Laboratory Foundation (614210302020 12103).

**Data Availability Statement:** Data is contained within the article.

**Conflicts of Interest:** The authors declare no conflicts of interest.

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
