# Peer review of "SM9 Identity-Based Encryption with Designated-Position Fuzzy Equality Test"

_electronics, doi:10.3390/electronics13071256_

Round 1
Reviewer 1 Report
Comments and Suggestions for Authors
The concept of identity-based encryption with designated-position fuzzy equality test (IBE-DFET) is truly innovative. By extending the functionality of equality tests to IBE frameworks, your work addresses critical privacy concerns while maintaining the flexibility of authorization. I appreciate the meticulous construction of the SM9-IBE-DFET scheme outlined in your paper. Your use of the Chinese national cryptography standard SM9-IBE algorithm not only enhances the practicality and security of the scheme but also facilitates its deployment. Your paper's thorough analysis of correctness and security, particularly under the F-OW-ID-CCA and IND-ID-CCA models, provides confidence in the robustness of your proposed scheme. This comprehensive approach to evaluating the scheme's effectiveness is commendable and adds significant value to your work. Your paper represents a significant advancement in cryptographic protocols, offering valuable insights into enhancing privacy and security in identity-based encryption schemes.
Reviewer 2 Report
Comments and Suggestions for Authors
This paper is novel enough and well-structured. It contains sound, rigorous correctness and security proof. However, there are some minor and major problems. The major problem is in the comparison with other works and the minor problem is with the English writing. A few comments appear in the following just to convey my meaning.
Abstract:
The abstract is too long.
"to decide that whether two ciphertexts" --> "to determine whether two ciphertexts"
"are generated from" --> "have been generated from"
"It should be noted that such primitive has not yet been introduced into" --> "Despite its proven efficiency in public key encryption, the existing literature does not incorporate PKE-DFET in"
Section 6.
The first sentence "We have conducted a visual comparison of our scheme with several existing ones [14, 516
37,46]," has been connected to the next sentence (by comma). I recommend "We have visually compared our scheme with several existing ones including [14], [37] and [46]."
[14], [37] and [46] need a more elaborative explanation before being compared with your own work.
Table 1 is all you have for comparison between your work and other works. You should interpret this table in more detail. For example, apparently, the only numerical measure you have for this comparison is "computational complexity". How do you define this measure? How do you calculate it? What is the value for your work and other works? How do they compare to each other?
In addition to performance, you may need numerical measures for comparison on the security. For example, you may want to provide some randomness test or another numerical analysis.
Comments on the Quality of English Language
I have mentioned only a few cases. However, there are several parts that require revision on the English writing.
Reviewer 3 Report
Comments and Suggestions for Authors
In this paper, the authors developed the identity-based encryption with designated position fuzzy equality test (IBE-DFET), which integrates the functionality of designated position fuzzy equality test into the IBE framework. The paper is well organized and has a good contribution. The following comments are for further consideration.
1. The authors should put the related work in a separate section after the introduction and merge the contributions with the introduction.
2. In the section that reviews related works and compares them to the proposed scheme, you have covered various important elements. To enhance the comprehensiveness and clarity of this review, it is suggested to add a table that summarize the key differences between the proposed scheme provided features and the related works in literature and also consider providing a brief preview of the comparison table at the beginning of this section to help readers anticipate the upcoming detailed comparisons.
3. Some abbreviations need to be explained before using them like F-OW-ID-CCA and IND-ID-CCA.
4. It is suggested that varying the number of features should be considered to assess its impact on computational and communication costs. Therefore, it is advisable to include an experiment that varies the number of features and includes it in the comparison with other methodologies in different figures. Addressing these concerns and making the necessary adjustments will enhance the clarity and completeness of the paper.
5. In physical reality, the time-efficiency may be the bottleneck for implementing an algorithm. However, this paper considers nothing about the algorithm time-efficiency, storage efficiency, etc.
6. In the experiment part, this paper only provides the straightforward result analysis. The author should provide insightful explanations on how proposed model works and why it can substantially improve the performance. For example, the authors should explain why the results indicate that your scheme achieves better performance than the others.
Comments on the Quality of English Language
The paper contains some grammatical errors and typos that have to be modified.
Round 2
Reviewer 2 Report
Comments and Suggestions for Authors
Thank you for the revision.
Reviewer 3 Report
Comments and Suggestions for Authors
The authors have addressed all of my comments, and I believe the paper is now ready for publication.